# Quantifying audio visual alcohol imagery in popular Indian films: a content analysis

Rohith Bhagawath,[1] Muralidhar M Kulkarni,[1] John Britton,[2] Jo Cranwell,[3] Monika Arora,[4,5] Gaurang P Nazar,[4,5] Somya Mullapudi,[1] Veena G Kamath[1]

[1]Community Medicine, Kasturba Medical College, Manipal Academy of Higher Education, Manipal, Karnataka, India
[2]Division of Epidemiology and Public Health, University of Nottingham, Nottingham, UK
[3]Tobacco Control Research Group, Department for Health, University of Bath, Bath, UK
[4]Health Related Information Dissemination Amongst Youth, HRIDAY, New Delhi, India
[5]Health Promotion Division, Public Health Foundation of India, New Delhi, India

**Correspondence to**
Professor John Britton;
j.britton@outlook.com

## ABSTRACT

**Objectives** Though exposure to alcohol imagery in films is a significant determinant of uptake and severity of alcohol consumption among young people, there is poor evidence regarding the content of alcohol imagery in films in low-income and middle-income countries. We have measured alcohol imagery content and branding in popular Indian films, in total and in relation to language and age rating.

**Design** In this observational study we measured alcohol imagery semiquantitatively using 5-minute interval coding. We coded each interval according to whether it contained alcohol imagery or brand appearances.

**Setting** India.

**Participants** None. Content analysis of a total of 30 national box office hit films over a period of 3 years from 2015 to 2017.

**Primary and secondary outcome measures** To assess alcohol imagery in Indian films and its distribution in relation to age and language rating has been determined.

**Results** The 30 films included 22 (73%) Hindi films and 8 (27%) in regional languages. Seven (23%) were rated suitable for viewing by all ages (U), and 23 (77%) rated as suitable for viewing by children subject to parental guidance for those aged under 12 (UA). Any alcohol imagery was seen in 97% of the films, with 195 of a total of 923 5-minute intervals, and actual alcohol use in 25 (83%) films, in 90 (10%) intervals. The occurrence of these and other categories of alcohol imagery was similar in U-rated and UA-rated films, and in Hindi and local language films. Episodes of alcohol branding occurred in 10 intervals in five films.

**Conclusion** Almost all films popular in India contain alcohol imagery, irrespective of age rating and language. Measures need to be undertaken to limit alcohol imagery in Indian films to protect the health of young people, and to monitor alcohol imagery in other social media platforms in future.

## BACKGROUND

Consumption of alcohol is a major avoidable cause of morbidity and mortality.[1] According to the WHO, there are 2.3 billion current alcohol users around the world,[2] and in 2016 alcohol consumption led to 2.8 million deaths.[3] Global alcohol consumption has increased by 38% in the last decade,[4]

### Strengths and limitations of this study

► Our study, to the best of our knowledge is the first of its kind to measure alcohol imagery in popular films in India.
► The study demonstrates that almost all popular Indian films contain alcohol imagery.
► We included only top 10 national box office hit films in the years 2015, 2016 and 2017 since coding films is time-consuming.
► Films in only four Indian languages figured in the box office hits; while films in several other regional languages did not.
► Although our study did not include any A-rated (adult) or S-rated (restricted to specialised audience) films, our findings related entirely to films classified by the Central Board of Film Certification as suitable for young people.

indicating that the burden of morbidity and mortality caused by alcohol is likely to grow.

Alcohol consumption is a significant public health problem in India.[5] Although rare among women, nearly one in three men now consume alcohol,[6] with an average intake of 18.3 L per year.[2] Although the mean age at which people begin consuming alcohol is 21 years,[7 8] 1.3% of children aged 10–17 report alcohol consumption,[9] and studies reveal that early onset of use of alcohol in India correlates with chronic heavy drinking patterns later in adult life.[7 8]

Research among adolescents in developed countries indicates that in addition to paid-for advertising, exposure to media imagery of alcohol products and alcohol consumption, particularly in films, plays an important role in promoting the uptake of alcohol consumption among young people.[10] However, while national film age-rating or classification systems typically include restrictions on alcohol imagery allowed in films rated suitable for viewing by children or young people, analysis of alcohol imagery in popular UK films demonstrates that alcohol content

**BMJ**

continues to occur frequently in films marketed to these groups.[11] To our knowledge, however there has to date been no evaluation of the extent to which popular films in India include alcohol imagery. This study was therefore carried out to quantify alcohol imagery, in relation to the age classification and language of the film, in a sample of the most popular films in India in the years 2015–2017.

## METHODS

We used Indian national box office ratings data to identify the top 10 Indian films by box office revenue in each of the years 2015, 2016 and 2017.[12] The Central Board of Film Certification (CBFC) age rating (comprising U (unrestricted public exhibition), UA (parental guidance for children below the age of 12 years), A (restricted to adults) and S (restricted to any special class of persons)) along with the genre and language of each film was noted. We used the validated 5-minute interval coding,[13 14] as described in previous studies, to code the presence of alcohol imagery in each interval as any use, actual use, implied use, other alcohol references and alcohol brand appearances. Actual use of alcohol was coded if an actor/actress in the film was shown consuming alcohol, while implied alcohol use typically involved verbal comments related to alcohol or non-verbal actions such as holding a glass or a bottle appearing to contain an alcoholic drink. Other alcohol references usually comprised the appearance of alcohol bottles or beer mugs. An occurrence in any of the above categories was coded as present or absent for each interval. Multiple appearances in the same category in the same interval were coded as a single event, and appearances in different categories as separate events. The same appearance transitioning into two or more intervals was coded as two or more events, as appropriate. Accuracy of results was ensured by two coders independently coding each of the movies with quality check done by two other coders. Any discrepancy was cleared after discussing with the investigators of the study. For alcohol branding we noted occurrences of clear, unambiguous alcohol branding on-screen, subgrouped as branding on a product used in a scene, branding on a product not used in a scene, branded merchandise, advertisements visible in scenes of alcohol content and any other alcohol-related advertisement. The total number of alcohol brands shown in were counted and listed.

### Patient and public involvement

No patients were involved in this study.

### Analysis

Frequencies and percentages were calculated for the language, age rating, alcohol usage and brand appearance. Mean and standard deviation was calculated for runtime of the film. Proportions of films and intervals were compared for alcohol content between films according to age classification and language using $\chi^2$ test.

## RESULTS

The 30 films included seven (23%) that were U-rated and 23 (77%) UA-rated, and 22 (73%) in Hindi and 8 (27%) in regional languages. The regional language included four Tamil, three Telugu and one Malayalam films. The mean (SD) film runtime of the 30 films was 151.2 (12.8) min, and their cumulative duration 4535 min, which we coded in 923 5-minute intervals. Of these, 229 (25%) were in U and 694 (75%) in UA films, and 665 (72%) in Hindi and 258 (28%) in regional language films. The list of the films with their language, genre, age rating and the number of intervals is provided in the online supplemental file 1. Any alcohol appearance was observed in 195 (21%) 5-minute intervals occurring in 29 (97%) of the films. The proportion of intervals containing alcohol was slightly lower in U-rated (38 intervals, 17%) than UA-rated (157 intervals, 23%) films, though this difference was statistically not significant (p>0.05); and the same in Hindi (140 intervals, 21%) and regional language (55 intervals, 21%) films (p>0.05).

Actual alcohol use appeared in 25 (83%) films, in 90 (10%) intervals. All seven U-rated films included actual alcohol use, which occurred in 19 (50%) of the 38 intervals including any alcohol imagery, with implied use and other alcohol references occurring in 11 (29%) and 8 (11%) of alcohol intervals, respectively. Actual alcohol use occurred in 18 (78%) of the 23 UA rated films in 71 (45%) of the 157 intervals including alcohol, while implied use and other alcohol references occurred in 55 (35%) and 37 (24%) of alcohol intervals, respectively. Actual alcohol use occurred in 18 (82%) of the 22 Hindi films, in 64 (46%) of the 140 intervals containing alcohol, with implied use and other alcohol references occurring in 54 (39%) and 25 (17%) of alcohol intervals, respectively; and in 7 of the 8 regional language films in 26 (47%) of the 55 intervals containing alcohol, with implied use and other alcohol references each occurring in 12 (22%) and 20 (40%) alcohol intervals, respectively, as shown in table 1.

Alcohol brand appearances occurred in 10 intervals in five films, one of which was U-rated and one in a regional language. The brands, Signature, Royal Stag, Carlsberg, Sula Wine, Old Smuggler, Kingfisher Premium, Kisset, Mcdonald, Brown Horse and Tuborg appeared once each, and Vat 69 twice.

## DISCUSSION

This study, to our knowledge the first of its kind in India, demonstrates that alcohol imagery occurred in almost all of this sample of films popular in India, all of which were classified by the CBFC in India as suitable either for unrestricted viewing by children (U), or for by children under the age of 12 with parental guidance (UA).[15] Most of the alcohol imagery comprised actual or implied use. In contrast with our earlier analysis of tobacco imagery in these films,[13] we found no evidence of greater alcohol content in films made in local languages. However, the proportion of both

**Table 1** Alcohol imagery based on language and Central Board of Film Certification rating of the films

| Type of alcohol imagery | Language | | | | | | | | Age rating | | | | | | | | All films | | | |
| | Hindi | | | | Regional | | | | U rated | | | | UA rated | | | | | | | |
| | 22 Films | | 665 Intervals | | 8 Films | | 258 Intervals | | 7 Films | | 229 Intervals | | 23 Films | | 694 Intervals | | 30 Films | | 923 Intervals | |
| | N | % | N | % | N | % | N | % | N | % | N | % | N | % | N | % | N | % | N | % |
| Any alcohol | 21 | 95.5 | 140 | 21.0 | 8 | 100.0 | 55 | 21.3 | 7 | 100.0 | 38 | 16.6 | 22 | 95.7 | 157 | 22.6 | 29 | 96.6 | 195 | 21.1 |
| Actual use | 18 | 81.8 | 64 | 9.6 | 7 | 87.5 | 26 | 10.1 | 7 | 100.0 | 19 | 8.3 | 18 | 78.3 | 71 | 10.2 | 25 | 83.3 | 90 | 9.8 |
| Implied use | 18 | 81.8 | 54 | 8.1 | 5 | 62.5 | 12 | 4.7 | 5 | 71.4 | 11 | 4.8 | 18 | 78.3 | 55 | 7.9 | 23 | 76.7 | 66 | 7.2 |
| Other references | 10 | 45.5 | 25 | 3.8 | 5 | 62.5 | 20 | 7.8 | 4 | 57.1 | 8 | 3.5 | 12 | 52.2 | 37 | 5.3 | 15 | 50.0 | 45 | 4.9 |
| Alcohol branding | 4 | 18.2 | 7 | 1.1 | 1 | 12.5 | 3 | 0.5 | 1 | 14.3 | 1 | 0.4 | 4 | 17.4 | 9 | 1.3 | 5 | 16.7 | 10 | 1.1 |

U, unrestricted public exhibition; UA, parental guidance for children below the age of 12 years.

U-rated and UA-rated films containing alcohol was similarly high and although the proportion of intervals containing alcohol was slightly lower in U-rated than in UA-rated films, this difference was not statistically significant. These findings suggest that films are a significant source of exposure to alcohol imagery for children and young people in India, and that the CBFC film classification system, which offers guidance on glorification or justification of drinking alcohol,[15] is not currently exerting appreciable influence on overall alcohol content.

Coding films is time-consuming so we confined our study to the most popular films by selecting the top 10 box office hit national films in each of 3 years. These did not include any A-rated or S-rated films, so our findings related entirely to films classified by the CBFC as suitable for young people. Films in India are made in 35 languages[16] while we coded only top 30 films that comprised of films in four languages. There is a need to code films in other languages to monitor alcohol imagery and take appropriate measures.

A limitation in the methodology is that we used 5-minute intervals as coding units and thereby multiple scenes, if present in a single interval might be underrated.[11] Coding can be done by methods such as 5-minute interval coding,[17] 1-minute interval coding,[18] using scene breaks to define intervals[19 20] or methods of continuous measurements,[21 22] with each having strengths and weaknesses. However, the 5-minute interval method has proved to be a sensitive means of capturing behavioural variation[23] and an effective semiquantitative method of measurement which balances accuracy with the logistic need to conduct highly time-consuming measurements efficiently. This method has been used in many studies and has been used in the present study also[24 25] to provide semiquantitative estimates of alcohol content. Our findings are consistent with reports from other countries: Hanewinkel et al, in a study in 2007,[26] reported alcohol imagery in 88% of 398 films coded, similar to another six-country study[10] published in 2014, where 86% of 655 films had alcohol content. The low occurrence of branding in the present study contrasts however with earlier work in the UK, which found frequent occurrences of American beer brands in popular UK films.[11]

Exposure of young people to alcohol imagery is unlikely to be limited to that in films watched in cinemas, because films are only one of many entertainment media through which children are exposed to alcohol imagery, and children consume a growing range of media, including social media, that have been shown to contain alcohol imagery.[27–29] However, films represent an important source of exposure, not only in terms of the cinema audiences they generate but also for the potentially wider audiences reached when films are shown on television. According to a Broadcast Audience Research Council report, young people represent 33% of total television viewership and films comprise more than half of that viewing in India.[30]

Film classification bodies such as the CBFC (and, for example, the British Board of Film Classification) tend to consider alcohol imagery as relevant to age

classification only if it involves glamorisation or glorification, but evidence on the effect of tobacco exposure in films suggests that these considerations are unimportant: all tobacco imagery promotes tobacco use, irrespective of who, how or in what other circumstances the product is consumed.[31] To date, evidence on exposure to alcohol imagery is less developed and further work is needed to establish the extent of this effect. However, the prudent approach to avoidable risks is to avoid them, and since the inclusion of alcohol imagery in films aimed at children is entirely avoidable, protecting children from future alcohol use and consequent problems justifies more rigorous controls on the alcohol content of films aimed at children and young people in India.

**Acknowledgements** The authors appreciate the assistance of the project staff of the Preventing smoking uptake study in procuring and coding films.

**Contributors** JB, MMK and RB conceptualised the study. The coding sheet was developed by RB, MMK, JC and GPN. Coding analysis was done by RB, MMK and VGK. Interpretation of data and drafting the manuscript was done by RB, MMK, VGK, SM and MA. Final approval of the version was read and approved by all authors.

**Funding** This work was supported by the Medical Research Council (grant number MR/P008933/1) of the UK under the Global Alliance for Chronic Lung diseases program.

**Competing interests** None declared.

**Patient consent for publication** Not required.

**Ethics approval** IEC: 893/2018

**Provenance and peer review** Not commissioned; externally peer reviewed.

**Data availability statement** Data are available upon reasonable request. Additional data on the alcohol content found in each film are available on request from rohit.bhagawath@manipal.edu.

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
