## [Reviewer comments · BMJ Open]

ARTICLE DETAILS

TITLE (PROVISIONAL)	Quantifying Audio Visual Alcohol Imagery in Popular Indian Films: A Content Analysis
AUTHORS	Kulkarni, Muralidhar; Britton, John; Cranwell, Jo; Arora, Monika; Nazar, Gaurang; Mullapudi, Somya; Kamath, Veena

VERSION 1 – REVIEW

REVIEWER	James Sargent Dartmouth Geisel School of Medicine USA
REVIEW RETURNED	23-Jul-2020

GENERAL COMMENTS	This is a content analysis of popular Indian films for alcohol. The authors employ a commonly used approach, in which five minute intervals are coded for use of alcohol and brand imagery. The study finds that alcohol appears frequently in Indian films and does not appear to be subject to ratings assessments, since they appear in equal frequency for films rated for all ages as for those rated for needing parental approval. General comments Alcohol in films is an exposure that, much like advertising, creates a basis for how children view this product and its consumption. Higher exposure increases the risk that children will try alcohol and also binge drink. Thus, tracking how often it appears in film is important. We know it appears often in western films but little is known about films from industries in the eastern part of the world. Bollywood is a large industry with great regional impact. The study is therefore important and was conducted with care. I have a few comments that would make the presentation stronger. CODING RELIABILITY The question always arises on content coding, how accurate was it. Accuracy is typically assessed by having two coders independently examine the same movie and then the investigators report inter-rater reliability. This was not reported here. It should be. Minor comments ABSTRACT Don't understand what "quantum" of alcohol imagery means in the outcomes section.
--

REVIEWER	Frank Houghton Limerick Institute of Technology, Ireland.
-----------------	--

REVIEW RETURNED	19-Oct-2020
-------------

GENERAL COMMENTS	A valid research question Abstract Page 6 line 11 the word 'but' seems unnecessary. Highlights Page7, line 57, 2017 in twice – did you mean 2016? Line 52 Page 10- I don't think the P for Patient needs to be in capitals Were Adult or Specialised films excluded – or were none top 10 box office hits? Alcohol was assessed in 5 minute segments. Why? Isn't one minute segments normal- please justify. Please write up the Chi-square calculations in standard reporting format. Don't discuss new material in the Discussion This is fine as far as it goes. Is it a short Report/ piece of Research Correspondence?
--

VERSION 1 – AUTHOR RESPONSE

Reviewer: 1

Reviewer Name: James Sargent

Institution and Country: Dartmouth Geisel School of Medicine, USA

Competing interests 1: None

Comments to the Author

This is a content analysis of popular Indian films for alcohol. The authors employ a commonly used approach, in which five minute intervals are coded for use of alcohol and brand imagery. The study finds that alcohol appears frequently in Indian films and does not appear to be subject to ratings assessments, since they appear in equal frequency for films rated for all ages as for those rated for needing parental approval.

General comments

Alcohol in films is an exposure that, much like advertising, creates a basis for how children view this product and its consumption. Higher exposure increases the risk that children will try alcohol and also binge drink. Thus, tracking how often it appears in film is important. We know it appears often in western films but little is known about films from industries in the eastern part of the world. Bollywood is a large industry with great regional impact. The study is therefore important and was conducted with care. I have a few comments that would make the presentation stronger.

Authors Reply: Thank you for highlighting the importance of the research paper.

Reviewers comment 1-CODING RELIABILITY The question always arises on content coding, how accurate was it. Accuracy is typically assessed by having two coders independently examine the same movie and then the investigators report inter-rater reliability. This was not reported here. It should be.

Authors Reply: We have incorporated the below statement in the revised manuscript and the text now reads as follows "Accuracy of results was ensured by two coders independently coding each of the movies with quality check done by two other coders. Any discrepancy was cleared after discussing with the investigators of the study." On Page 6, line 17 .

Minor comments

Reviewers comment 2- ABSTRACT

Don't understand what "quantum" of alcohol imagery means in the outcomes section.

Authors Reply: Sorry for the lack of clarity. "Quantum" indicates the magnitude of the variable of interest (alcohol imagery). We have edited the same as "to assess" on Page 3, line 13.

Reviewer: 2

Reviewer Name: Frank Houghton

Institution and Country: Limerick Institute of Technology, Ireland.

Comments to the Author

Competing interests: None

A valid research question

Reviewers comment 1 - Abstract Page 6 line 11 the word 'but' seems unnecessary.

Authors Reply: We have edited in the manuscript accordingly in Page 3 line 3.

Reviewers comment 2- Highlights Page7, line 57, 2017 in twice – did you mean 2016?

Authors Reply: Thank you for pointing the error. Yes, it is 2016, and has been incorporated accordingly in the manuscript on Page 4, line 8.

Reviewers comment 3- Line 52 Page 10- I don't think the P for Patient needs to be in capitals

Authors Reply: We have edited the same on Page 7, Line 1.

Reviewers comment 4- Were Adult or Specialised films excluded – or were none top 10 box office hits?

Authors Reply: The top 10 box office hits did not include the Adult or Specialised films and the same has been clarified on Page 9, Line 3.

Reviewers comment 5- Alcohol was assessed in 5 minute segments. Why? Isn't one minute segments normal- please justify.

Authors Reply: We have redrafted the discussion to deal with this problem in more detail, adding a paragraph on page 9 which reads: The semi-quantitative interval coding method we used employs 5 minute segments has been widely used in previous work by our group. We have made changes in the manuscript on Page 9, Line 9 which reads

"Coding can be done by methods such as 5 min interval coding¹⁷, 1 min interval coding¹⁸, using scene breaks to define intervals^{19,20} or methods of continuous measurements^{21,22}, with each having strengths and weaknesses. However, the 5 min interval method has proved to be a sensitive means of capturing behavioural variation²³ and an effective semi-quantitative method of measurement which balances accuracy with the logistic need to conduct highly time-consuming measurements efficiently. This method has been used in many studies and has been used in the present study also.^{24,25} to provide semi-quantitative estimates of alcohol content. Our findings are consistent with reports from other countries: Hanewinkel et al, in a study in 2007 ²⁶, reported alcohol imagery in 88% of 398 films coded, similar to another six-country study 10 published in 2014, wherein 86% of 655 films had alcohol content. The low occurrence of branding in the present study contrasts however with earlier work in the UK, which found frequent occurrences of American beer brands in popular UK films¹¹."

Reviewer's comment 6- Please write up the Chi-square calculations in standard reporting format.
Authors Reply: As the chi square value was not significant we modified the manuscript accordingly in standard chi square reporting format ($p>0.05$).

Reviewer's comment 7- Don't discuss new material in the Discussion

Authors Reply: The typographical error in Page 8 line 15 might have given a wrong impression of a new result, we apologise and have now rectified the reference number to superscript. Hope this clarifies.

The phrase removed was, "YouTube, Facebook, Instagram, Netflix and Amazon Prime" and the line reads as "Exposure of young people to alcohol imagery is unlikely to be limited to that in films watched in cinemas, because films are only one of many entertainment media through which children are exposed to alcohol imagery, and children consume a growing range of media, including social media, that have been shown to contain alcohol imagery" In page 9 line 24.

Reviewer's comment 8 - This is fine as far as it goes. Is it a short Report/ piece of Research Correspondence?

Authors Reply: We have submitted it to the journal as an original research article.

VERSION 2 – REVIEW

REVIEWER	Frank Houghton Limerick Institute of Technology, Ireland
REVIEW RETURNED	05-Jan-2021
GENERAL COMMENTS	This is a useful addition to the literature. The pervasiveness of alcohol imagery in our enviroPAGE 9 To our knowledge, however there here 4 has to date been no evaluation of the extent to which popular films in India include 5 alcohol imagery.